# HART: Efficient Visual Generation with Hybrid Autoregressive Transformer

**Haotian Tang**[1*]  **Yecheng Wu**[1,3*]  **Shang Yang**[1]  **Enze Xie**[2]  **Junsong Chen**[2]
**Junyu Chen**[1,3]  **Zhuoyang Zhang**[1]  **Han Cai**[2]  **Yao Lu**[2]  **Song Han**[1,2]
MIT[1]  NVIDIA[2]  Tsinghua University[3]
https://hanlab.mit.edu/projects/hart

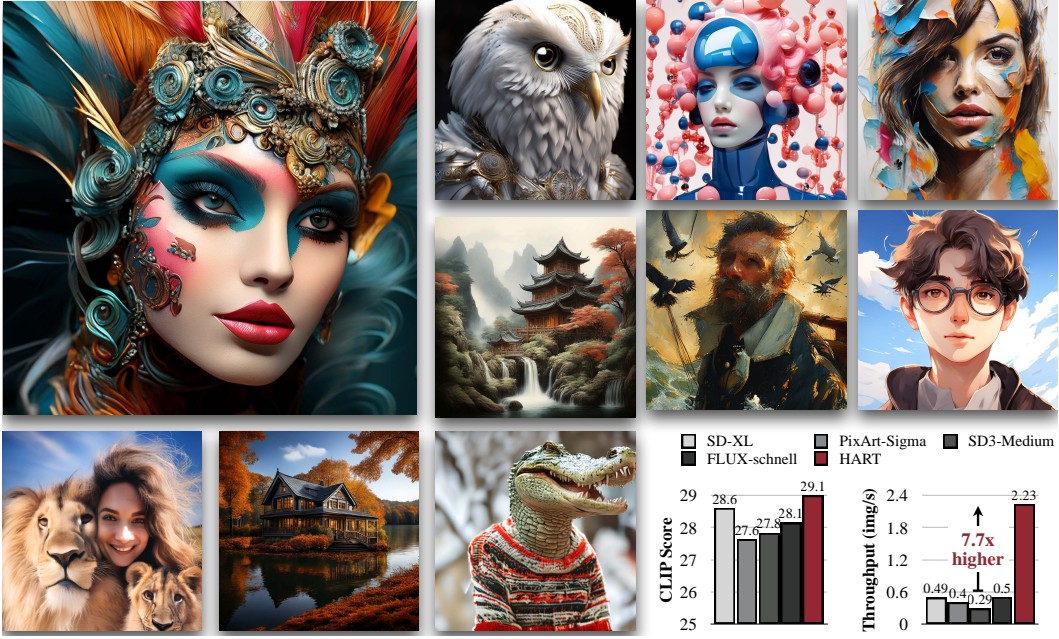

Figure 1: **HART** is an early autoregressive model that can directly generate **1024×1024** images with quality comparable to diffusion models, while offering significantly improved efficiency. It achieves **4.5-7.7×** higher throughput, **3.1-5.9×** lower latency (measured on A100), and **6.9-13.4×** lower MACs compared to state-of-the-art diffusion models. Check out our online demo and video.

## ABSTRACT

We introduce *Hybrid Autoregressive Transformer* (HART), an autoregressive (AR) visual generation model capable of directly generating 1024×1024 images, rivaling diffusion models in image generation quality. Existing AR models face limitations due to the poor image reconstruction quality of their discrete tokenizers and the prohibitive training costs associated with generating 1024px images. To address these challenges, we present the *hybrid tokenizer*, which decomposes the continuous latents from the autoencoder into two components: discrete tokens representing the big picture and *continuous* tokens representing the residual components that cannot be represented by the discrete tokens. The discrete component is modeled by a *scalable-resolution* discrete AR model, while the continuous component is learned with a lightweight *residual diffusion* module with only 37M parameters. Compared with the discrete-only VAR tokenizer, our hybrid approach improves reconstruction FID from **2.11** to **0.30** on MJHQ-30K, leading to a **31%** generation FID improvement from **7.85** to **5.38**. HART also outperforms state-of-the-art diffusion models in both FID and CLIP score, with **4.5-7.7×** higher throughput and **6.9-13.4×** lower MACs. Our code is open sourced at https://github.com/mit-han-lab/hart.

---

*indicates equal contribution. Part of the work was done when Haotian Tang and Shang Yang were summer interns at NVIDIA. Correspondence to: {kentang,songhan}@mit.edu.

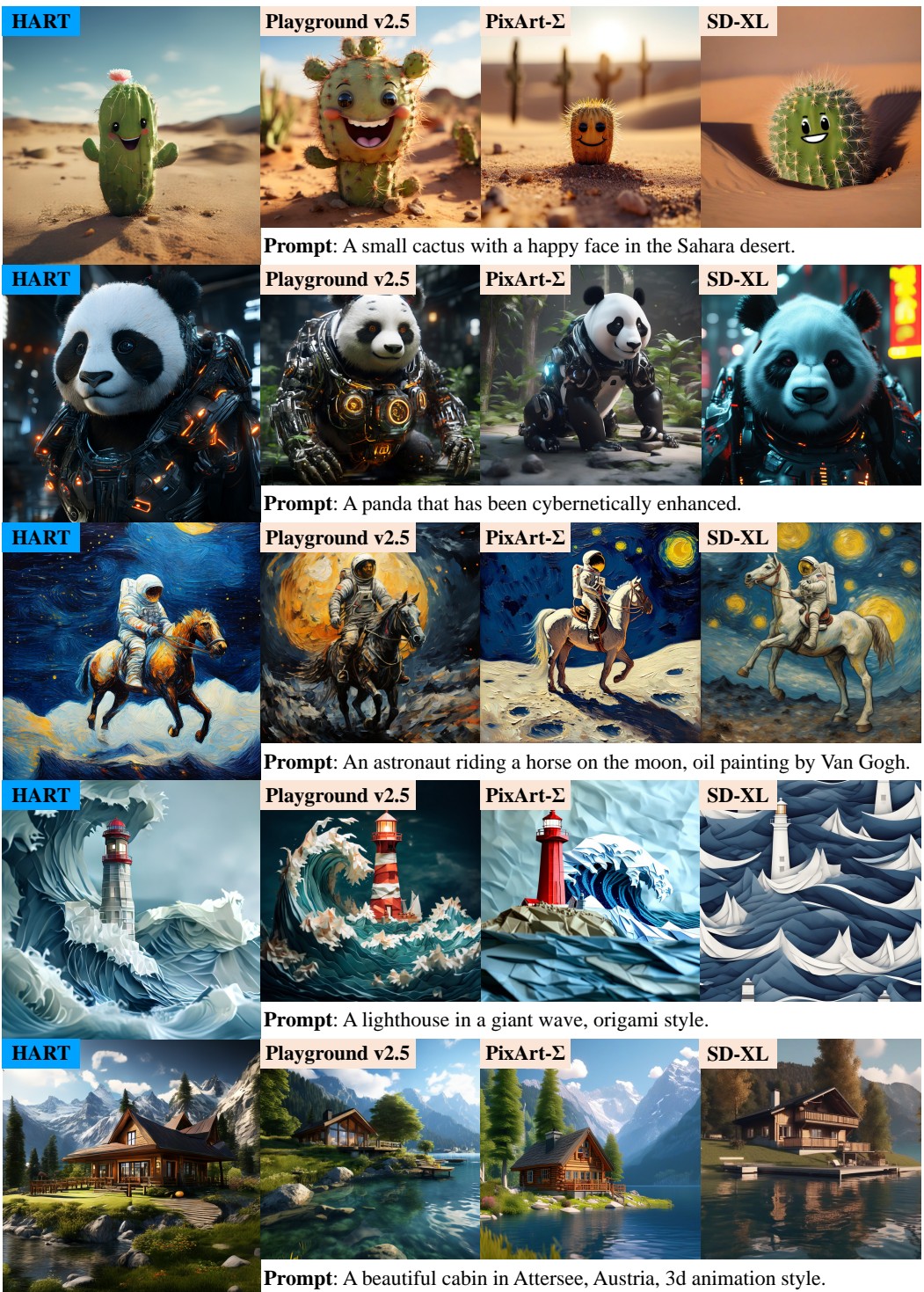

Figure 2: HART generates 1024px images with quality comparable to state-of-the-art diffusion models such as Playground v2.5 (Li et al., 2024a), PixArt-Σ (Chen et al., 2024a), and SDXL (Podell et al., 2023) while being **4.6-5.6×** faster.

# 1 INTRODUCTION

The rapid advancement of large language models (LLMs) is pushing artificial intelligence into a new era. At the core of LLMs are autoregressive (AR) models, which have gained popularity due to their generality and versatility. These models typically predict the next token in a sequence based on the previous tokens as input. While originating from natural language processing, autoregressive models have also recently been adopted for visual generation tasks. These approaches utilize a visual tokenizer to convert images from pixel space into discrete visual tokens through vector quantization (VQ) (Van Den Oord et al., 2017). These visual tokens are then processed in the same manner as language tokens. Benefiting from techniques proven successful in the LLM field, autoregressive visual generation methods have demonstrated their effectiveness in diverse tasks, including text-to-image generation, text-to-video generation, and image editing (Van Den Oord et al., 2017; Esser et al., 2021; Chang et al., 2022; Yu et al., 2022b; Kondratyuk et al., 2023; Tian et al., 2024). Autoregressive image generation models have also demonstrated significant potential for building unified visual language models (Gemini Team, Google, 2023; OpenAI, 2024), such as Emu3 (Emu3 Team, BAAI, 2024), VILA-U (Wu et al., 2024), and Show-o (Xie et al., 2024).

Concurrently, another major trend in visual generation from Ho et al. (2020); Rombach et al. (2022); Chen et al. (2024a); BlackForest Labs (2024) has centered on diffusion models. These models employ a progressive denoising process, beginning with random Gaussian noise. Diffusion models achieve better generation quality compared with autoregressive models, but they can be computationally expensive to deploy: even with an efficient DPM-Solver sampler from Lu et al. (2022), it still takes DiT-XL/2 (Peebles & Xie, 2023) 20 denoising steps to generate an image, which translates to **86.2T** MACs at 1024×1024 resolution. In contrast, generating a comparable image using a similarly sized AR model capable of predicting multiple tokens in parallel (Tian et al., 2024) requires only **10.1T** MACs at the same resolution, which is **8.5× less** computationally intensive.

This paper addresses the following question: ***Can we develop an autoregressive model that matches the visual generation quality of diffusion models while still being significantly faster?***

Currently, visual generation AR models lag behind diffusion models in two key aspects:

1. Discrete tokenizers in AR models exhibit significantly poorer reconstruction capabilities compared to the continuous tokenizers used by diffusion models. Consequently, AR models have a lower generation upper bound and struggle to accurately model fine image details.

2. Diffusion models excel in high-resolution image synthesis, but no existing AR model can directly generate 1024×1024 images.

To address these challenges, we introduce **HART** (Hybrid Autoregressive Transformer) for efficient high-resolution visual synthesis. HART bridges the reconstruction performance gap between discrete tokenizers in AR models and continuous tokenizers in diffusion models through *hybrid tokenization*. The hybrid tokenizer decomposes the continuous latent output of the autoencoder into two components: one as the *sum of discrete latents* derived from a VAR tokenizer (Tian et al., 2024), and the other as the *continuous residual*, representing the information that cannot be captured by discrete tokens. The discrete tokens captures the big picture, while continuous residual tokens focus on fine details (Figure 3). These two latents are then modeled by our *hybrid transformer*: the discrete latents are handled by a *scalable-resolution* VAR transformer, while the continuous latents are predicted by a lightweight *residual diffusion* module with **5%** parameter and **10%** runtime overhead.

HART achieves significant improvements in both image tokenization and generation over its discrete-only baseline. Compared with VAR, it reduces the reconstruction FID from **2.11** to **0.30** at 1024×1024 resolution on MJHQ-30K (Li et al., 2024a), enabling HART to lower the 1024px generation FID on the same dataset from **7.85** to **5.38** (a **31%** relative improvement). Furthermore, we demonstrate that HART achieves up to a **7.8%** improvement in FID over VAR for class-conditioned generation on ImageNet (Deng et al., 2009). HART also outperforms MAR on this task with **13×** higher throughput.

Notably, HART closely matches the quality of state-of-the-art diffusion models in multiple text-to-image generation metrics. Simultaneously, HART achieves **3.1-5.9×** faster inference latency, **4.5-7.7×** higher throughput, and requires **6.9-13.4×** less computation compared with diffusion models.

Figure 3: HART synergizes discrete and continuous tokens. The discrete tokens capture the overall image structure, while the fine details (*e.g.* eyes, eyebrows and hair) are reflected in the *residual tokens*, which is modeled by *residual diffusion* (introduced in Section 3.2).

## 2 RELATED WORK

Visual generation has become a key focus in machine learning research. Initial work by Kingma & Welling (2013) introduced variational autoencoders (VAEs) for image synthesis. Subsequently, Goodfellow et al. (2014) proposed generative adversarial networks (GANs), which were further improved by Brock et al. (2018); Karras et al. (2019); Kang et al. (2023).

**Diffusion models** from Ho et al. (2020); Nichol & Dhariwal (2021); Dhariwal & Nichol (2021); Ramesh et al. (2022); Betker et al. (2023) have emerged as the state-of-the-art approach for generating high-quality images after VAE and GAN. The latent diffusion model from Rombach et al. (2022); Podell et al. (2023) applies U-Net to denoise the Gaussian latent input, and is succeeded by DiT from Peebles & Xie (2023) and U-ViT from Bao et al. (2023) which replaces the U-Net with transformers. Chen et al. (2023; 2024b;a) scale up DiTs to text-to-image (T2I) generation. Concurrently, Kolors Team (2024); Ma et al. (2024a); Li et al. (2024a) further scaled up T2I diffusion models to billions of parameters. Recent research from Esser et al. (2024); Auraflow Team (2024); BlackForest Labs (2024) also explored rectified flow for fast sampling.

**Autoregressive models** pioneered by Chen et al. (2020) generate images as pixel sequences, rather than denoising an entire latent feature map simultaneously. Early research VQVAE and VQGAN from Van Den Oord et al. (2017); Esser et al. (2021) quantize image patches into discrete tokens and employ a decoder-only transformer to predict these image tokens, analogous to language modeling. VQGAN was subsequently enhanced in several aspects: Yu et al. (2022a) improved its autoencoder modeling, Chang et al. (2022); Yu et al. (2023a); Li et al. (2023) increased its sampling speed with MaskGIT, while Mentzer et al. (2023), Yu et al. (2023b), and Yu et al. (2024) enhanced its tokenization performance and efficiency. Lee et al. (2022) introduced residual quantization to reduce tokenization error. Building on this, Tian et al. (2024) developed VAR, which innovatively transformed next-token prediction in RQVAE to next-scale prediction, significantly improving sampling speed. There were also efforts that scaled up autoregressive models to text-conditioned visual generation: Ramesh et al. (2021); Ding et al. (2021; 2022); Liu et al. (2024); Sun et al. (2024); Crowson et al. (2022); Gafni et al. (2022); Emu3 Team, BAAI (2024) were T2I generation methods based on VQGAN, and Chang et al. (2023); Villegas et al. (2022); Kondratyuk et al. (2023); Xie et al. (2024) extended MaskGIT. STAR, VAR-CLIP and ControlVAR from Ma et al. (2024b); Zhang et al. (2024); Li et al. (2024c) were extensions of VAR.

**Hybrid models** represent a new class of visual generative models that synergize discrete and continuous image modeling approaches. GIVT from Tschannen et al. (2023) predicted continuous visual tokens with autoregressive models while VQ-Diffusion from Gu et al. (2022) extended diffusion to discrete latents. MAR from Li et al. (2024b) and DisCo-Diff from Xu et al. (2024) conditioned a diffusion model with autoregressive prior. This idea was also concurrently explored in visual language models by Ge et al. (2024); Jin et al. (2023). Transfusion (Zhou et al., 2024) fuses DiT and LLM into a single model, and is natively capable of multi-modal generation.

## 3 HART: HYBRID AUTOREGRESSIVE TRANSFORMER

We introduce Hybrid Autoregressive Transformer (HART) for image generation. Unlike all existing generative models that operate exclusively on either discrete or continuous latent spaces, HART

models both discrete and continuous tokens with a unified transformer. The key factors enabling this are a *hybrid tokenizer* and *residual diffusion*.

## 3.1 HYBRID VISUAL TOKENIZATION

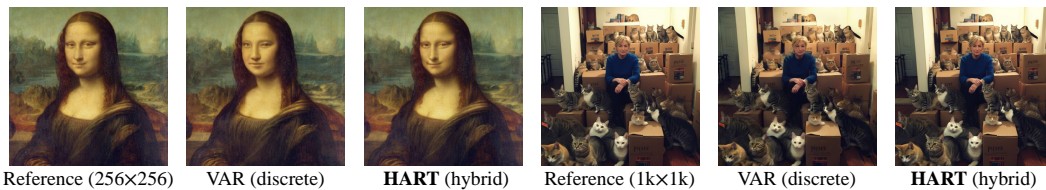

Reference (256×256)  VAR (discrete)  **HART** (hybrid)  Reference (1k×1k)  VAR (discrete)  **HART** (hybrid)

Figure 4: **Reconstruction quality comparison between VAR and HART tokenizers.** The discrete tokenizer employed by VAR will lose some details or have some distortion during the reconstruction, which is solved by hybrid tokenization in HART. Please zoom in for details in 1k images.

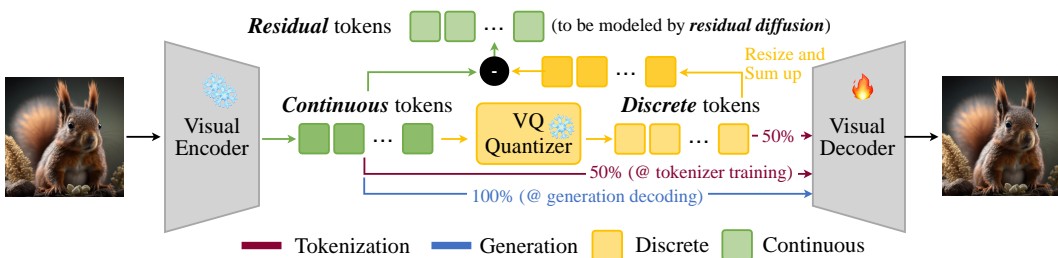

Figure 5: Unlike conventional image tokenizers that decode either continuous *or* discrete latents, the **hybrid tokenizer** in HART is trained to decode both continuous *and* discrete tokens. At inference time, we only decode continuous tokens, which are the sum of discrete tokens and residual tokens. The residual tokens will be modeled by residual diffusion (introduced in Figure 6).

Conventional autoregressive visual generation encodes images into discrete tokens using trained tokenizers. These tokens map to entries in a vector-quantized (VQ) codebook and can reconstruct the original images from the VQ tokens. This approach transforms text-to-image generation into a sequence-to-sequence problem, where a decoder-only transformer, or LLM, predicts image tokens from text input. The tokenizer's reconstruction quality sets the upper limit for image generation. Constrained by their finite vocabulary codebooks, discrete tokenizers often struggle to faithfully reconstruct images with intricate, high-frequency details such as human faces, as in Figure 4.

**Hybrid tokenization.** We introduce our hybrid tokenizer in Figure 5. The primary goal of hybrid tokenization is to enable the decoding of *continuous features* during generation, thereby overcoming the poor generation upper bound imposed by finite VQ codebooks. We begin with a CNN-based visual encoder that transforms the input image into continuous visual tokens in the latent space. These tokens are then quantized into discrete tokens across multiple scales, following VAR (Tian et al., 2024). The multi-scale vector quantization process results in a difference between the accumulated discrete features and the original continuous visual features, which can not be accurately represented using VQ codebook elements. We term this difference *residual tokens*, which are subsequently modeled by *residual diffusion*, as detailed in Section 3.2.

**Alternating training.** To train our hybrid tokenizer, we begin by initializing the visual encoder, quantizer (*i.e.*, codebook), and visual decoder from a pretrained discrete VAR tokenizer. We then freeze the visual encoder and quantizer, allowing only the visual decoder to be trained. During each training step, we randomly choose with equal probability (50%) whether to provide the decoder with discrete or continuous visual tokens for reconstructing the input image. Specifically, when the continuous path is selected (lower red path in Figure 5), it bypasses the VQ quantizer, effectively turning the model into a conventional continuous autoencoder. Otherwise, if the discrete path is selected (upper red path in Figure 5), we are essentially training a standard VQ tokenizer. Empirical results show that the HART tokenizer achieves comparable continuous rFID (*i.e.*, reconstruction FID

when the continuous path is activated) to the SDXL tokenizer (Podell et al., 2023), while its discrete rFID matches the performance of the original VQ tokenizer. As a result, the generation upper bound of HART remains consistent with state-of-the-art diffusion models. This alternating training strategy also ensures that the continuous and discrete latents remain sufficiently similar from the decoder's perspective, facilitating easier modeling of continuous latents.

**Discussions.** It's important to note that low rFID does not necessarily indicate better generation FID. The alternating approach in training our hybrid tokenizer is crucial for high-quality generation. In Section 4.3, we demonstrate that other methods, such as using separate decoders for continuous and discrete tokens, may achieve similar continuous reconstruction FID but significantly compromise generation FID. Furthermore, the next subsection explains why autoregressive methods utilizing continuous tokenizers, like MAR (Li et al., 2024b), are less efficient than HART.

## 3.2 HYBRID AUTOREGRESSIVE MODELING WITH RESIDUAL DIFFUSION

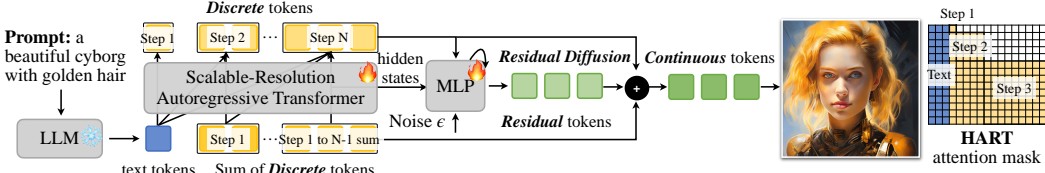

Figure 6: **HART** is an efficient hybrid autoregressive image generation framework. It decomposes continuous image tokens into two components: 1) a series of *discrete* tokens modeled by a *scalable-resolution* (up to 1024px) autoregressive transformer, and 2) *residual* tokens modeled by a lightweight *residual diffusion* (37M parameters and 8 steps) module. The final image representation is the sum of these two components.

Hybrid tokenization offers superior rFID and a better generation upper bound compared to discrete tokenization. We introduce HART (Figure 6) to efficiently translate this improved upper bound into real enhancements in generation quality. HART models the continuous image tokens as the sum of two components: (1) *discrete* tokens, modeled by a *scalable-resolution* autoregressive transformer, and (2) *residual* tokens, fitted by an efficient residual diffusion process.

**Scalable-resolution autoregressive transformer.** Our discrete token modeling extends VAR to text-to-image generation and improves scalability at higher resolutions. HART concatenates text tokens with visual tokens during training, in contrast to VAR which use a single class token. The text tokens are visible to all visual tokens, as in Figure 6 (right). Our approach is 25% more parameter-efficient than STAR (Ma et al., 2024b)'s cross-attention method (Chen et al., 2023).

Unlike Parti, the only prior AR-based method achieving 1024px generation through super-resolution with Imagen (Saharia et al., 2022), HART directly generates 1024px images with a single model. To mitigate the $O(n^4)$ training cost for high-resolution AR transformers, we finetune from pre-trained low-resolution checkpoints. We convert all absolute position embeddings (PEs) in VAR to interpolation-compatible relative embeddings, including step (indicating the resolution each token belongs to) and token index embeddings. We utilize sinusoidal PE for step embeddings, which naturally accommodates varying sampling steps in 256/512px (10 steps) and 1024px (14 steps) generation. For token index embeddings, we implement a hybrid approach: 1D rotary embeddings for text tokens and 2D rotary embeddings (Sun et al., 2024; Ma et al., 2024a; Wang et al., 2024) for visual tokens. The position indices of visual tokens directly continue from those of text tokens. We found these relative embeddings significantly accelerates HART convergence at higher resolutions.

**Residual diffusion.** We employ diffusion to model residual tokens, given their continuous nature. Similar to MAR (Li et al., 2024b), we believe that a full DiT is unnecessary for learning this residual. Instead, a lightweight (37M parameters) *residual diffusion* MLP would be sufficient. This MLP is conditioned on the last layer hidden states from our scalable-resolution AR transformer, as well as the discrete tokens predicted in the last VAR sampling step.

Despite similar denoising MLP model architectures, HART differs fundamentally from MAR. While MAR predicts *full* continuous tokens, HART models *residual* tokens—a crucial distinction for effi-

cient diffusion modeling. Although both trained with a 1000-step noise schedule, HART achieves optimal quality with just **8** sampling steps at inference, compared to MAR's 30-50, resulting in a **4-6×** reduction in diffusion module overhead. This demonstrates that HART's residual tokens are significantly easier to learn than MAR's full tokens.

**Other differences.** The AR transformers in HART and MAR differ significantly in their formulation. MAR's AR transformer generates only conditions for its diffusion MLP, lacking a discrete codebook for token generation. In contrast, HART's AR transformer produces both discrete tokens and diffusion conditions. These discrete tokens can be decoded into meaningful, albeit less detailed, images using our hybrid tokenizer design (Figure 3). This approach reduces the burden on residual diffusion, which only needs to model fine details rather than the overall image structure. Furthermore, HART supports KV caching for faster inference, significantly reducing computational costs. MAR's transformer, based on MaskGIT (Chang et al., 2022), lacks this capability.

In contrast to other representative AR+diffusion methods such as LaVIT (Jin et al., 2023) and SEED-X (Ge et al., 2024), which employ complete diffusion models (1B parameters, 20 steps) for full continuous tokens, HART provides significant efficiency gains through the use of a tiny diffusion MLP (37M parameters, 8 steps) that models only residual tokens.

### 3.3 Efficiency Enhancements

While our scalable-resolution AR transformer and residual diffusion designs are crucial for high-quality, high-resolution image generation, they inevitably introduce inference and training overhead. We address these efficiency challenges in this section.

**Training.** Naively adding the residual diffusion module incurs both computational and memory overhead during training. To address this, we found that discarding 80% of the tokens (on average) in the final step and applying supervision only to the remaining tokens during training does not degrade performance. This approach accelerates training by **1.4×** at 512px and **1.9×** at 1024px, while also reducing training memory usage by $1.1\times$. In the appendix, we explain the effectiveness of this method by demonstrating that the attention pattern in our autoregressive transformer is mostly local. Consequently, although token subsampling during training may compromise global interactions between tokens, it has small impact on attention calculation.

**Inference.** For inference, we observed that relative position embeddings introduced multiple memory-bound GPU kernel calls, in contrast to the single call required for absolute position embeddings in VAR (Tian et al., 2024). To optimize performance, we fused these computations into two kernels: one for sinusoidal calculation and another for rotary embedding. This optimization resulted in a 7% improvement in end-to-end execution time. Additionally, fusing all operations in RMSNorm into a single GPU kernel also improved total runtime by 10%.

## 4 Experiments

In this section, we evaluate HART's performance in tokenization and generation. For generation, we present both text-to-image and class-conditioned image generation results.

### 4.1 Setup

**Models.** For class-conditioned image generation models, we follow VAR (Tian et al., 2024) to construct HART models with varying parameter sizes in the AR transformer: 600M, 1B, and 2B. The diffusion MLP contains an additional 37M parameters. We replace VAR's attention and FFN blocks with Llama-style (Touvron et al., 2023) building blocks. For text-conditioned image generation, we start with the 1B model and remove all AdaLN (Peebles & Xie, 2023) layers, resulting in a 30% reduction in parameters. We employ Qwen2-1.5B (Yang et al., 2024) as our text encoder and follow LI-DiT (Ma et al., 2024a) to reformat user prompts.

**Evaluation and Datasets.** We evaluate HART on ImageNet (Deng et al., 2009) for class-conditioned image generation, and on MJHQ-30K (Li et al., 2024a), GenEval (Ghosh et al., 2024), and DPG-Bench (Hu et al., 2024) for text-to-image generation. The HART tokenizer is trained on OpenImages (Kuznetsova et al., 2020). For HART transformer training, we utilize ImageNet, JourneyDB (Pan et al., 2023), and internal MidJourney-style synthetic data. All text-to-image generation

| Type | Model | #Params | Resolution | MJHQ-30K | | GenEval | DPG-Bench |
|------|-------|---------|------------|----------|---------|---------|-----------|
| | | | | FID↓ | CLIP-Score↑ | Overall↑ | Average↑ |
| Diff. | SD v2.1 | 860M | 768×768 | 26.96 | 25.90 | 0.50 | 68.09 |
| Diff. | SD-XL | 2.6B | 1024×1024 | 8.76 | 28.60 | 0.55 | 74.65 |
| Diff. | PixArt-$\alpha$ | 630M | 512×512 | 6.14 | 27.55 | 0.48 | 71.11 |
| Diff. | PixArt-$\Sigma$ | 630M | 1024×1024 | 6.34 | 27.62 | 0.52 | 79.46 |
| Diff. | Playground v2.5 | 2B | 1024×1024 | 6.84 | 29.39 | 0.56 | 76.75 |
| Diff. | SD3-medium | 2B | 1024×1024 | 11.92 | 27.83 | 0.62 | 85.80 |
| AR | LlamaGen | 775M | 512×512 | 25.59 | 23.03 | 0.32 | 65.16 |
| AR | Show-o | 1.3B | 256×256 | 14.99 | 27.02 | 0.53 | 67.48 |
| AR | HART | 732M | 512×512 | 5.22 | 29.01 | 0.56 | 80.72 |
| | | | 1024×1024 | 5.38 | 29.09 | 0.56 | 80.89 |

Table 2: The performance of HART on MJHQ-30K, GenEval and DPG-Bench benchmarks. We compare HART with open-source diffusion models and autoregressive models. Results demonstrate that HART can achieve comparable performance to state-of-the-art diffusion models with <1B parameters, surpassing prior autoregressive models by a large margin.

| Model | #Params | #Steps | 512×512 | | | 1024×1024 | | |
|-------|---------|--------|---------|---------|---------|---------|---------|---------|
| | | | Latency (s) | Throughput (image/s) | MACs (T) | Latency (s) | Throughput (image/s) | MACs (T) |
| SDXL | 2.6B | 20 | 1.4 | 2.1 | 30.7 | 2.3 | 0.49 | 120 |
| | | 40 | 2.5 | 1.4 | 61.4 | 4.3 | 0.25 | 239 |
| PixArt-$\Sigma$ | 630M | 20 | 1.2 | 1.7 | 21.7 | 2.7 | 0.4 | 86.2 |
| Playground v2.5 | 2B | 20 | – | – | – | 2.3 | 0.49 | 120 |
| | | 50 | – | – | – | 5.3 | 0.21 | 239 |
| SD3-medium | 2B | 28 | 1.4 | 1.1 | 51.4 | 4.4 | 0.29 | 168 |
| LlamaGen | 775M | 1024 | 37.7 | 0.4 | 1.5 | – | – | – |
| HART | 732M | 10 | **0.3** | **10.6** | **3.2** | – | – | – |
| | | 14 | – | – | – | **0.75** | **2.23** | **12.5** |

Table 3: Compared to state-of-the-art diffusion models, HART achieves **5.0-9.6×** higher throughput and **4.0-4.7×** lower latency at 512×512 resolution. At 1024×1024 resolution, it demonstrates **4.5-7.7×** higher throughput and **3.1-5.9×** lower latency.

data are recaptioned using VILA1.5-13B (Lin et al., 2024). We measure all quality and efficiency metrics using open-source models with recommended sampling parameters as released by their authors. Latency and throughput (batch=8) measurements are conducted on NVIDIA A100.

## 4.2 MAIN RESULTS

**Hybrid tokenization.** We evaluate the HART hybrid tokenizer on ImageNet and MJHQ-30K, two datasets not observed during training. As shown in Table 1, our hybrid tokenization offers significant advantages over discrete tokenization, reducing the 1024px rFID from 2.11 to **0.30**. This matches the performance level of the SDXL tokenizer, indicating that the generation upper bound of HART is comparable to that of diffusion models. The discrete rFID of our hybrid tokenizer also ensures that the discrete tokens still capture the majority of image structure, so that the residual tokens remain easily learnable.

| Method | MJHQ-30K rFID↓ | | | ImageNet rFID↓ | |
|--------|------|------|--------|------|------|
| | 256px | 512px | 1024px | 256px | 512px |
| VAR | 1.42 | 1.19 | 2.11 | 0.92 | 0.58 |
| SDXL | 1.08 | 0.54 | 0.27 | 0.69 | 0.28 |
| Ours (dis.) | 1.70 | 1.64 | 1.09 | 1.04 | 0.89 |
| Ours | 0.78 | 0.67 | 0.30 | 0.41 | 0.33 |

Table 1: HART significantly outperforms VAR and matches SDXL tokenizer performance on MJHQ-30K and ImageNet datasets.

**Text-to-image generation.** We present quantitative text-to-image generation results in Table 2. On MJHQ-30K, our method achieved superior FID compared to all diffusion models. HART also demonstrates better image-text alignment than the 3.6× larger SD-XL (Podell et al., 2023), as indicated by the CLIP score on the same dataset. On GenEval and DPG-Bench, HART achieves results comparable to diffusion models with <2B parameters. Importantly, HART achieves this generation quality at a significantly lower computational cost. As shown in Table 3, HART achieves a **9.3×** higher throughput compared to SD3-medium (Esser et al., 2024) at 512×512 resolution. For 1024×1024 generation, HART achieves at least **3.1×** lower latency than state-of-the-art diffusion

| Type | Model | FID↓ | IS↑ | #Params | #Step (AR.) | #Step (diff.) | MACs | Time (s) |
|------|-------|------|-----|---------|-------------|---------------|------|----------|
| Diff. | DiT-XL/2 | 2.27 | 278.2 | 675M | – | 250 | 57.2T | 113 |
| AR | VAR-$d$20 | 2.57 | 302.6 | 600M | 10 | – | 412G | 1.3 |
| AR | VAR-$d$24 | 2.09 | 312.9 | 1.0B | 10 | – | 709G | 1.7 |
| AR | VAR-$d$30 | 1.92 | 323.1 | 2.0B | 10 | – | 1.4T | 2.6 |
| AR | MAR-B | 2.31 | 281.7 | 208M | 64 | 100 | 7.0T | 26.1 |
| AR | MAR-L | 1.78 | 296.0 | 479M | 64 | 100 | 16.0T | 34.9 |
| AR | HART-$d$20 | 2.39 | 316.4 | 649M | 10 | 8 | 579G | 1.5 |
| AR | HART-$d$24 | 2.00 | 331.5 | 1.0B | 10 | 8 | 858G | 1.9 |
| AR | HART-$d$30 | 1.77 | 330.3 | 2.0B | 10 | 8 | 1.5T | 2.7 |

Table 4: HART achieves better class-conditioned image generation results compared to MAR (Li et al., 2024b) with **10.7×** lower MACs and **12.9×** faster runtime. It also offers **7.8%** FID reduction with **4%** runtime overhead compared with VAR (Tian et al., 2024). Time: bs=64 on A100.

models. Compared to the similarly sized PixArt-$\Sigma$ (Chen et al., 2024a), our method achieves **3.6×** faster latency and **5.6×** higher throughput, which closely aligns with the theoretical **5.8×** reduction in MACs. Compared to SDXL, HART not only achieves superior quality across all benchmarks in Table 2, but also demonstrates **3.1×** lower latency and **4.5×** higher throughput.

**Class-conditioned generation.** Table 4 presents our class-to-image generation results. HART outperforms MAR-L (Li et al., 2024b) in terms of FID and inception score, while requiring **10.7×** fewer MACs and achieving **12.9×** lower latency. Across all model sizes, HART demonstrates a **4.3-7.8%** improvement in FID and consistent enhancement in inception score. For larger models ($d \geq 24$), the residual diffusion overhead accounts for only **4-11%** of the total runtime. HART also compares favorably to DiT-XL/2 (Peebles & Xie, 2023), with our largest model being **3.3×** faster, even when DiT employs a 20-step sampler.

### 4.3 Ablation Studies and Analysis

| Depth | Res. tokens | FID↓ | IS↑ | Time (s) |
|-------|-------------|------|-----|----------|
| 20 | ✗ | 2.67 | 297.3 | 1.3 |
| 20 | ✓ | **2.39** | **316.4** | 1.5 |
| 24 | ✗ | 2.23 | 312.7 | 1.7 |
| 24 | ✓ | **2.00** | **331.5** | 1.9 |
| 30 | ✗ | 2.00 | 311.8 | 2.5 |
| 30 | ✓ | **1.77** | **330.3** | 2.7 |

| Resolution | Res. tokens | FID↓ | CLIP↑ | Time (s) |
|------------|-------------|------|-------|----------|
| 256px | ✗ | 6.11 | 27.96 | 2.23 |
| 256px | ✓ | **5.52** | **28.03** | 2.42 |
| 512px | ✗ | 6.29 | 28.91 | 5.62 |
| 512px | ✓ | **5.22** | **29.01** | 6.04 |
| 1024px | ✗ | 5.73 | 29.08 | 25.9 |
| 1024px* | ✗ | 7.85 | 28.85 | 25.9 |
| 1024px | ✓ | **5.38** | **29.09** | 28.7 |

Table 5: HART learns *residual tokens*, which enhance conditioned image generation as evidenced by improvements in FID, inception score, and CLIP score. The HART-VAR results are obtained by omitting residual diffusion from the full HART model. Left: class-to-image, right: text-to-image, *: results obtained using the official VAR quantizer.

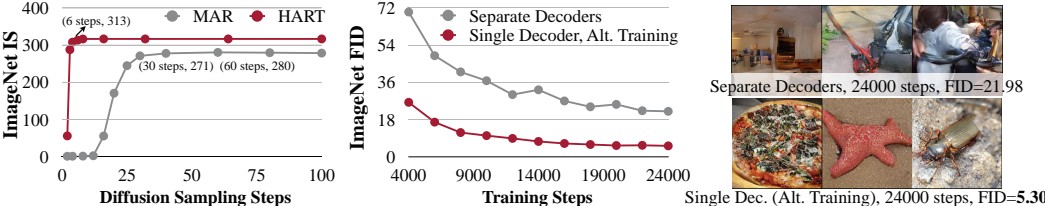

Figure 7: **Left**: *residual* tokens in HART are much easier to learn than *full* tokens in MAR. **Middle/Right**: Despite achieving similar reconstruction FID, single decoder with alternating training enables faster and better generation convergence.

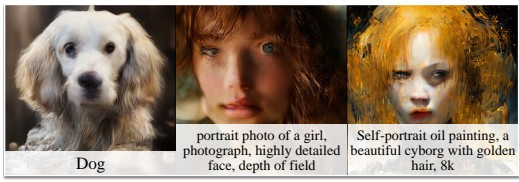 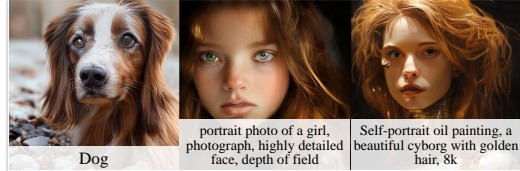

Official VAR, 256px→512px, 4000 steps    **HART** (scalable-resolution), 256px→512px, 4000 steps

Figure 8: **Scalable-resolution transformer** accelerates convergence when finetuning HART at higher resolution thanks to relative position embeddings that supports resolution interpolation.

We evaluate the key design choices in HART by examining: the effectiveness and efficiency of residual diffusion, the impact of our alternating training strategy for the hybrid tokenizer, and the importance of the scalable-resolution AR transformer.

**Residual diffusion: effectiveness.** Table 5 demonstrates the effectiveness of learning residual tokens in HART. For ImageNet 256×256 generation, residual diffusion yields a **10-14%** improvement in FID and up to a **6.4%** increase in inception score compared to the baseline model, HART-VAR. We constructed HART-VAR using the publicly available VAR codebase, which resulted in slightly lower discrete-only performance than reported in Tian et al. (2024). For text-conditioned generation on MJHQ-30K, FID improved by **11.1%** at 256px and **17.0%** at 512px. At 1024px, where the original VAR tokenizer shows poor reconstruction performance (Table 1), HART achieves a **31%** FID improvement. Even compared to the stronger discrete-only HART tokenizer, residual diffusion still offers a **6.1%** FID improvement. Figure 3 visualizes the residual tokens, illustrating how residual diffusion enhances discrete tokens with high-resolution details.

**Residual diffusion: efficiency.** Figure 7 (left) demonstrates that HART's approach of learning *residual* tokens is significantly more efficient than MAR's method of learning *full* tokens. Notably, HART achieves a higher inception score with just 3 diffusion sampling steps compared to MAR's 60 denoising steps, resulting in a **20×** reduction in runtime for continuous token learning.

**Alternating training in hybrid tokenizer.** We explored various strategies to train the hybrid tokenizer while maintaining similar continuous rFID. Figure 7 (middle and right) compares our current approach (single decoder with alternating training) to using separate decoders for continuous and discrete latents, with the discrete decoder frozen. Our design offers faster, better convergence for class-conditioned image generation. Alternative strategies, such as finetuning the entire hybrid tokenizer from a pretrained continuous tokenizer or decoding only continuous latents during training, are proven to be as bad as the separate decoder solution.

**Scalable-resolution transformer.** Lastly, Figure 8 illustrates that substituting all absolute PEs in VAR with relative PEs significantly enhances convergence when fine-tuning HART at higher resolutions from pretrained low-resolution checkpoints. Given that the token count in HART increases by 4× (resulting in a 16× increase in attention computation) when the output image resolution doubles, this accelerated convergence is crucial for maintaining manageable training costs.

# 5 CONCLUSION

We introduce HART (Hybrid Autoregressive Transformer), the first autoregressive model capable of directly generating 1024×1024 images from text prompts without super-resolution. HART achieves quality comparable to diffusion models while being **3.1-5.9×** faster and offering **4.5-7.7×** higher throughput. Our key insight lies in the decomposition of continuous image latents through *hybrid tokenization*, producing discrete tokens that capture the overall structure and *residual* tokens that refine image details. We model the discrete tokens using a *scalable-resolution* AR transformer, while a lightweight *residual diffusion* module with just 37M parameters and 8 sampling steps learns the residual tokens. We believe HART will catalyze new research into modeling both discrete and continuous tokens for sequence-based visual generation.

ETHICS STATEMENTS: DISCUSSION OF POTENTIAL MISUSE OF HART

Misuse of generative AI for creating NSFW (not safe for work) content remains a significant concern in the community. To enhance safety, we have integrated HART with ShieldGemma-2B (Zeng et al., 2024), a powerful LLM-based safety check model. In our implementation, user prompts are first evaluated by the safety check model to determine whether they contain NSFW content, including harmful, abusive, hateful, sexually explicit, or otherwise inappropriate material targeting individuals or protected groups. If the prompt is deemed appropriate, it proceeds to HART for image generation. Otherwise, the prompt is rejected and replaced with our default prompt ("A red heart"). Through extensive testing, we have confirmed that our safety check model effectively filters out NSFW prompts under strict thresholds, ensuring our pipeline does not generate harmful AI content.

ACKNOWLEDGMENTS

We thank MIT-IBM Watson AI Lab, MIT and Amazon Science Hub, MIT AI Hardware Program, National Science Foundation for supporting this research. We thank NVIDIA for donating the DGX server. We would like to also thank Tianhong Li from MIT, Lijun Yu from Google DeepMind, Kaiwen Zha from MIT and Yunhao Fang from UCSD for helpful technical discussions, and Paul Palei, Mike Hobbs, Chris Hill and Michel Erb from MIT for helping us set up the online demo and maintain the servers.

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

# A APPENDIX

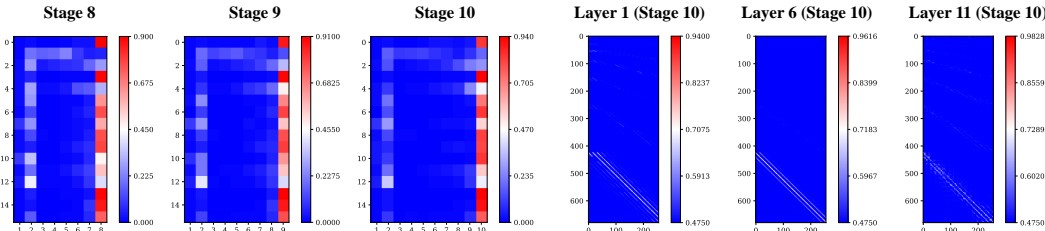

(a) The VAR attention in HART follows the stage-wise "sink + local" pattern    (b) Within the last stage, the attention pattern is local

Figure 9: **Left**: The VAR attention in HART exhibits a *sink + local* pattern: for stages 8-10 visualized here, attention scores concentrate in the *most recent two stages* and the *first three stages*, akin to StreamingLLM. **Right**: Within the final stage, the attention score distribution is predominantly *local*. Note: For clearer visualization, we apply a sigmoid function to the attention scores in the rightmost three subfigures.

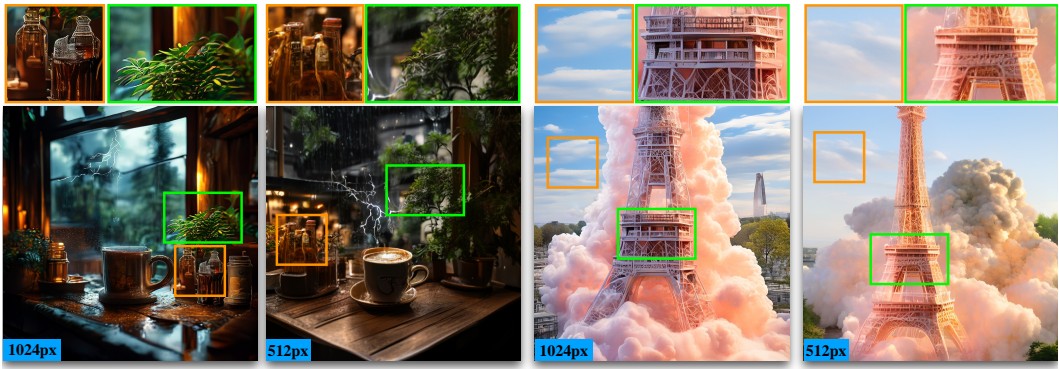

**Prompt:** A 3D render of a coffee mug placed on a window sill during a stormy day. The storm outside the window is reflected in the coffee, with miniature lightning bolts and turbulent waves seen inside the mug. The room is dimly lit, adding to the dramatic atmosphere. A minimap diorama of a cafe adorned with indoor plants. Wooden beams crisscross above, and a cold brew station stands out with tiny bottles and glasses.

**Prompt:** Eiffel Tower was Made up of more than 2 million translucent straws to look like a cloud, with the bell tower at the top of the building, Michel installed huge foam-making machines in the forest to blow huge amounts of unpredictable wet clouds in the building's classic architecture.

Figure 10: Direct high-resolution (1024×1024) image generation yields significantly more detailed results compared to low-resolution (512×512) generation.

## A.1 ATTENTION PATTERN ANALYSIS

We visualize the attention patterns of a pretrained VAR (Tian et al., 2024) in Figure 9. Our empirical analysis reveals that for each VAR stage (i.e., sampling step), the attention score is predominantly concentrated on three key areas: the current stage, the preceding stage, and the initial three stages.

Within the current stage, where the attention score is highest, we further examine the spatial attention map, as depicted in the rightmost three subfigures of Figure 9. Interestingly, despite the VAR attention mechanism allowing all tokens within the last stage to interact, the attention map exhibits a surprisingly localized pattern: each token primarily attends to its immediate neighbors, similar to convolution operations.

This observation has important implications. Even when we significantly reduce the number of tokens in the last stage during training (by up to 80%), the fundamental attention pattern remains intact due to the limited global interaction between tokens. This explains why the partial supervision approach during training (discussed in Section 3.3) does not compromise generation quality.

We have also empirically verified that explicitly restricting attention patterns to the first 3 stages plus 2 local stages during training does not impact final results. Consequently, implementing a sparse attention kernel to further accelerate training is feasible, which we leave as a future direction.

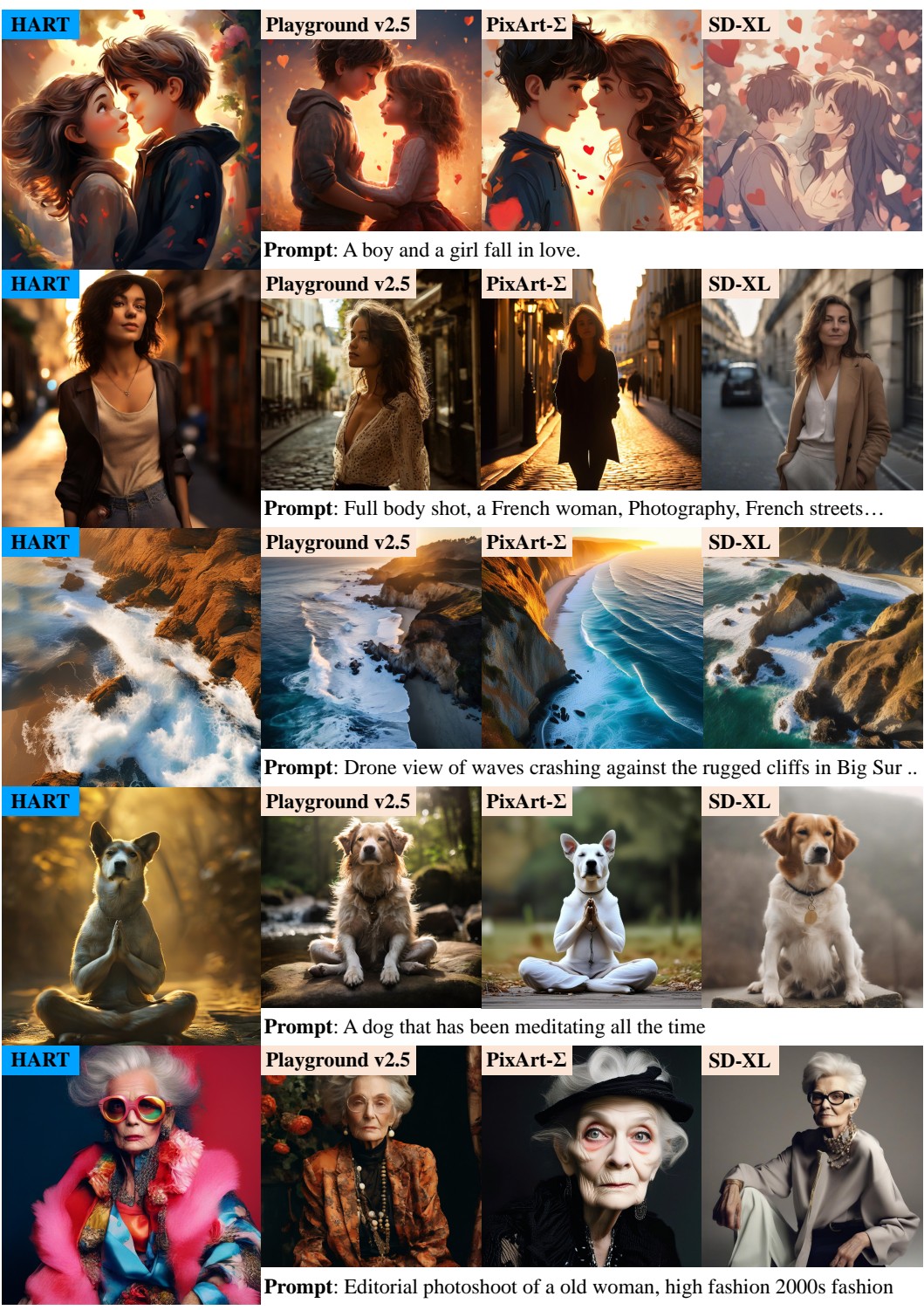

Figure 11: Additional 1024×1024 text-to-image generation results with HART. Full prompt for example 2: Full body shot, a French woman, Photography, French Streets background, backlighting, rim light, Fujifilm. Full prompt for example 3: Drone view of waves crashing against the rugged cliffs along Big Sur's Garay Point beach. The crashing blue waters create white-tipped waves, while the golden light of the setting sun illuminates the rocky shore.

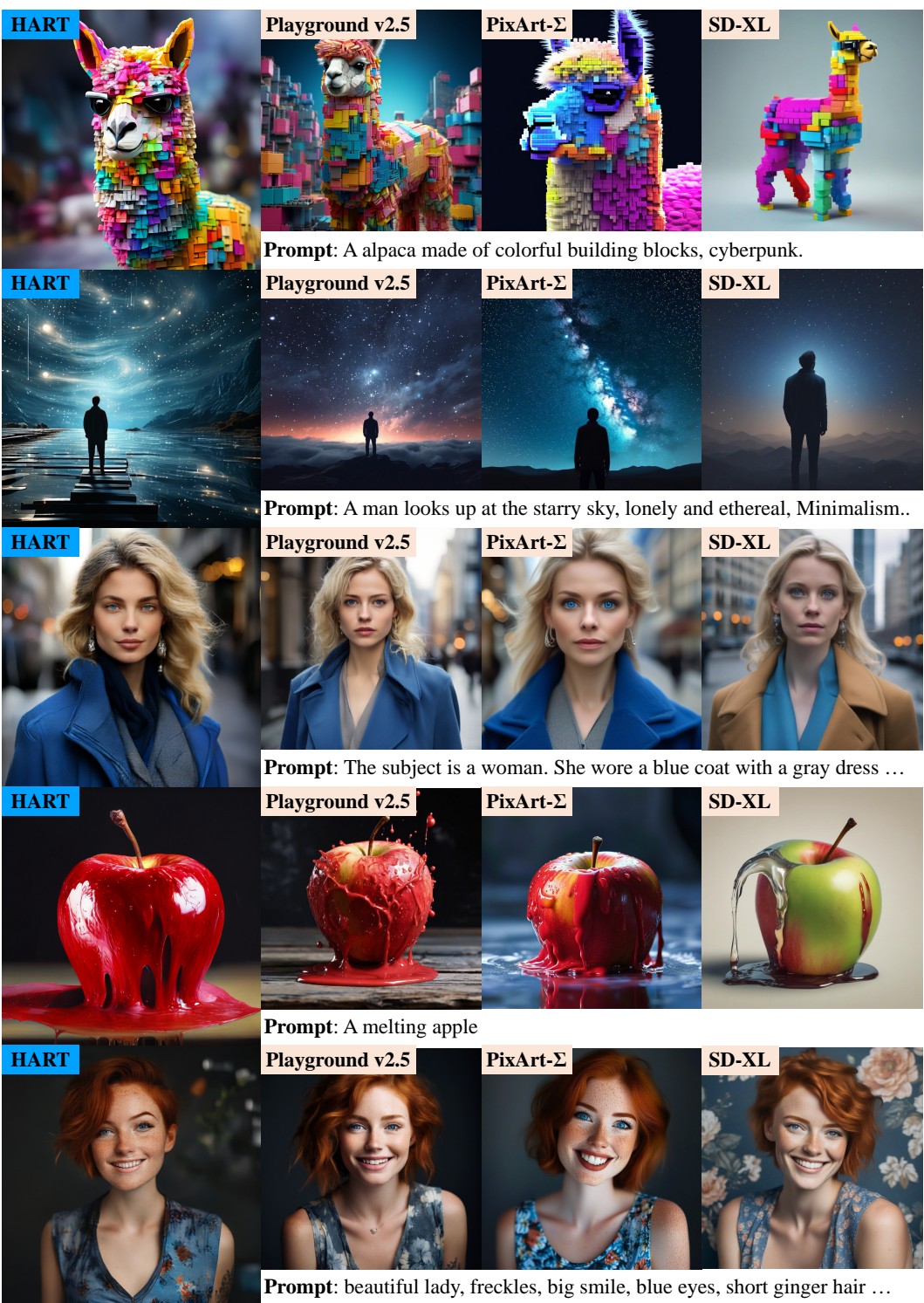

Figure 12: Additional 1024×1024 text-to-image generation results with HART. Full prompt for example 2: 8k uhd A man looks up at the starry sky, lonely and ethereal, Minimalism, Chaotic composition Op Art. Full prompt for example 3: A close-up photo of a person. The subject is a woman. She wore a blue coat with a gray dress underneath. She has blue eyes and blond hair, and wears a pair of earrings. Behind are blurred city buildings and streets. Full prompt for example 5: beautiful lady, freckles, big smile, blue eyes, short ginger hair, dark makeup, wearing a floral blue vest top, soft light, dark grey background.

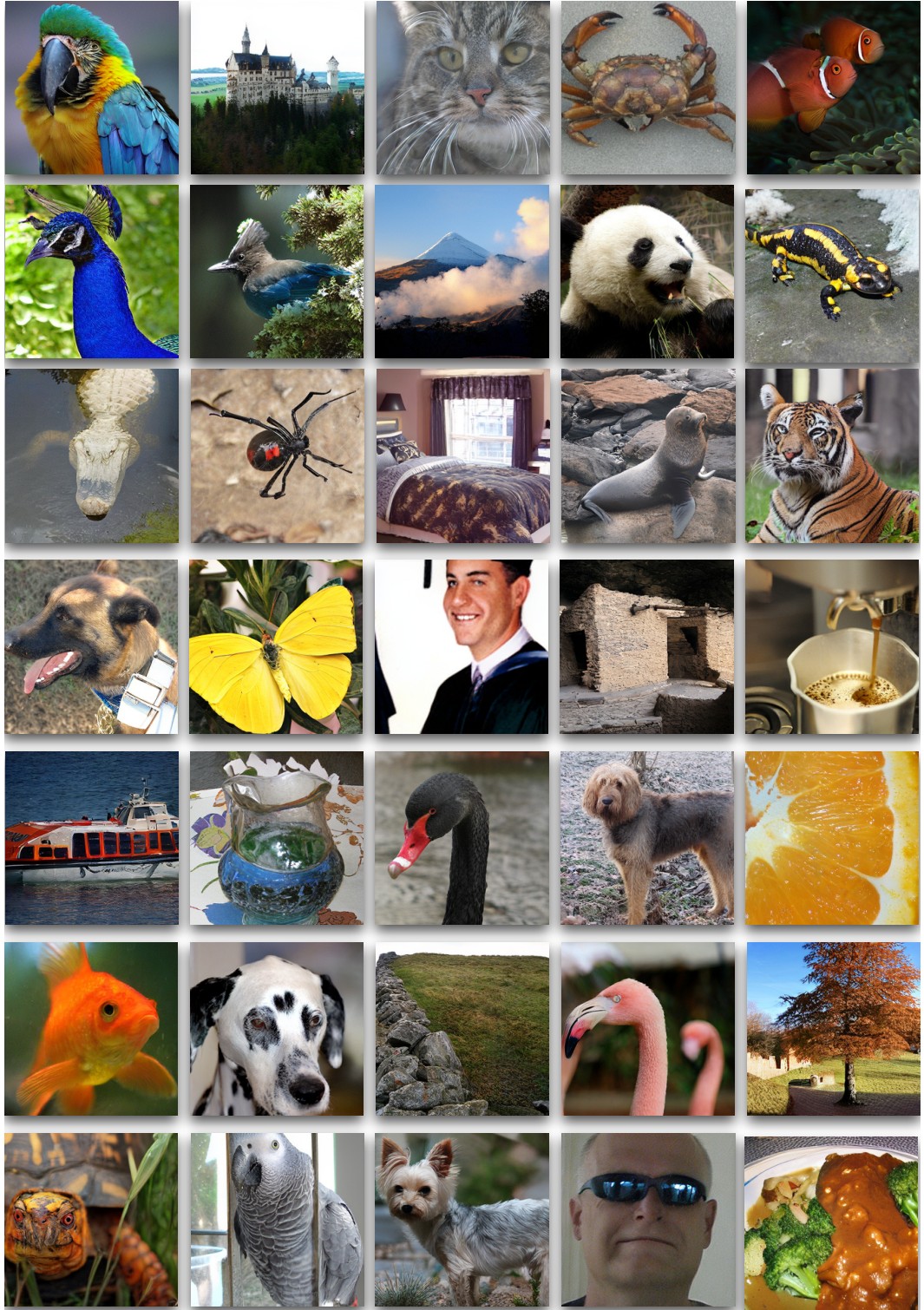

Figure 13: 256×256 class-conditional generation results from HART on ImageNet (Deng et al., 2009).

## A.2 MORE VISUALIZATIONS

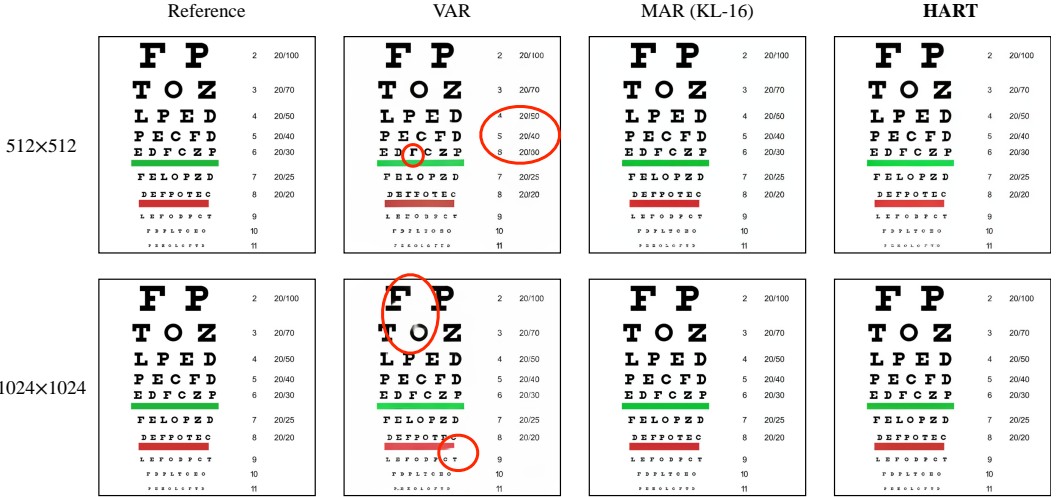

Figure 14: Additional image reconstruction comparison among VAR (discrete), MAR (KL-16, continuous) and HART (hybrid) tokenizers.

Figure 10 demonstrates the significant impact of direct synthesis at 1024×1024 resolution: the 1024px generated images exhibit substantially more details compared to their 512px counterparts. Figures 11 and 12 demonstrate text-conditioned generation. HART produce these images with comparable quality to diffusion models, while offering up to **7.7×** higher throughput. In Figure 13, we also showcase additional visualizations of HART-generated images for class-conditioned generation. Finally, Figure 14 presents additional image reconstruction results for various tokenizers. The HART tokenizer demonstrates reconstruction performance comparable to MAR's continuous tokenizer, while significantly outperforming VAR's discrete tokenizer.

