# OpenReview forum: "HART: Efficient Visual Generation with Hybrid Autoregressive Transformer"
_ICLR.cc/2025/Conference — ICLR 2025 Poster_

### Official Review · Reviewer_a2Ay · 2024-10-29

**Soundness:** 3
**Presentation:** 3
**Contribution:** 3
**Rating:** 6
**Confidence:** 4

**Summary:**

This paper presents an efficient and high-resolution autoregressive visual generation model with a hybrid tokenizer. The hybrid tokenizer includes a discrete tokenizer and a continuous tokenizer. The discrete tokenizer that is widely used in existing AR models is applied to model image structure, while the continuous tokenizer is used to represent residual image details. To model these image token priors, this paper also introduces a scalable discrete AR model and a light-weight residual diffusion model. Based on above modules, the proposed method beats VAR in rFID and outperforms sota diffusion models in both FID and CLIP score.

**Strengths:**

+ This paper proposes a hybrid tokenizer, addressing the poor image reconstruction quality of existing discrete tokenizer. Also, the hybrid tokenizer is simple and easy to implement.
+ To model tokens generated by the hybrid tokenizer, this paper introduce a scalable-resolution discrete AR model and a light-weight residual diffusion model. Based on these modules, the proposed method can generate 1024px image efficiently.
+ The authors conduct exhaustive experiments to validate the proposed method. Compared with existing AR and diffusion models, the proposed method achieves superior performance on text-to-image and class-conditioned image generation tasks.

**Weaknesses:**

- The condition of residual diffusion. As shown in line 320~321, the condition consists of last layer hidden states from AR transformer and the discrete tokens predicted in the last sampling step. To my knowledge, the discrete tokens predicted in the last sampling step only contain residual image details. It is not enough to predict the residual tokens. Thus, the hidden states are important. Does the hidden states come from all sampling steps?
- Efficiency enhancements (Training). In line 346~349, the fist sentence is about the overhead of residual diffusion module. But the following solution (discarding 80% tokens) is used to mitigate training overhead of VAR, which is also illustrated in Appendix A.1. In my opinion, there is a mismatch between these two parts.
- Alternating training in hybrid tokenizer. To my knowledge, with this training scheme, visual decoder is robust to the error of predicted tokens. For generation model, the error is large in the early training. Thus, the decoder with this scheme performs better than others, as shown in Figure 7 (middle and right). Does the great advantage of alternating training exist after generation model training?

**Questions:**

- The illustration of Figure 6. First, the arrow direction of scalable-resolution autoregressive transformer is wrong. Second, the last layer hidden states are not figured out in Figure 6. Third, the HART attention mask (right) does not contain much information. The attention mechanism of each stage is not easy to obtain for the readers.

---

> ### Author Response · Authors · 2024-11-23
> **Author response**
>
> **Q1: Clarification of condition in residual diffusion.**
>
> We clarify that the hidden states of the last layer contain information from all previous sampling steps. This is because the input at each sampling step is the cumulative feature map from all prior steps (see Figure 6). As a result, the residual diffusion module has the full context necessary to predict the final continuous residual tokens: discrete tokens from steps 1 to N-1, as well as the newly predicted discrete tokens from step N.
>
> **Q2: Description on efficiency improvement.**
>
> We have revised our description. Our original intention was to convey that naively adding the residual diffusion module introduces overhead in both training time and memory. However, our partial supervision strategy significantly accelerates training and reduces memory footprint.
>
> **Q3: The great advantage of alternating training still exists in the end of training.**
>
> We train the HART model with a separate-decoder hybrid tokenizer until convergence and benchmark its performance against the original HART model.
>
> | Method | rFID | gFID |
> | -------- | -------- | -------- |
> | Separate decoders |   0.43   |  5.48   |
> | HART (alternating training) |   **0.41**    |  **2.39**   |
>
> Clearly, the hybrid tokenizer trained with our alternating strategy maintains a huge performance advantage even after the hybrid transformer has converged.
>
> **Q4: Updating Figure 6.**
>
> Thanks for your helpful comments. We've updated Figure 6 accordingly.

---

> > ### Comment · Reviewer_a2Ay · 2024-11-24
> > **Thanks authors for the responses**
> >
> > My concerns have been solved. I prefer to keep my score as 6.

---

> > > ### Author Response · Authors · 2024-11-24
> > >
> > > We sincerely thank the reviewer for their prompt response and positive evaluation once again!

---

### Official Review · Reviewer_y4uu · 2024-11-01

**Soundness:** 3
**Presentation:** 3
**Contribution:** 3
**Rating:** 8
**Confidence:** 4

**Summary:**

This paper introduces HART, an innovative autoregressive transformer designed for efficient visual generation. It employs a hybrid token space to represent images, with the discrete component modeled by a VAR model and the continuous component modeled by a compact diffusion model. Extensive experiments demonstrate that HART outperforms previous baselines in both image quality and efficiency.

**Strengths:**

1. The paper is well-written and easy to follow.
2. The idea of using a compact diffusion model to model the residual continuous token is both simple and effective. I believe this design will offer valuable insights to the community.
3. The experiments are carefully designed. The results and conclusions are convincing.

**Weaknesses:**

The paper notes that a lower rFID does not necessarily indicate a better gFID, which is also supported by the ablation experiments. However, I feel that there is a lack of analysis regarding this phenomenon. For instance, it would be beneficial to provide insights into which design elements are crucial for obtaining tokens that achieve both good rFID and gFID simultaneously, and the other designs lead to a bad gFID.

**Questions:**

see Weakness.

---

> ### Author Response · Authors · 2024-11-23
> **Author response**
>
> **Q1: Analysis on the correlation between rFID and gFID in HART.**
>
> We sincerely thank the reviewer for the positive comments.
>
> An important principle for achieving both good rFID and gFID simultaneously is ensuring that the decoder used in generation can achieve strong discrete rFID. We assume that continuous rFID will generally perform well across different designs. We apply this principle to explain the two designs discussed in the paper:
>
> 1. **HART's Tokenizer**: A single decoder is used for generation. Its discrete rFID is competitive (Table 1), and the gFID is also strong.
> 2. **Separate Decoder Approach** (Figure 7, middle and right): Two decoders are used—one for continuous features and one for discrete features. The discrete decoder is frozen, and the continuous decoder is trained solely on continuous features. During generation, the **continuous decoder** is used, but its **discrete rFID is poor** (since it is not trained on discrete tokens), which results in a lower gFID.
>
> We have found this principle useful in explaining other unsuccessful designs, such as using a single decoder but only fine-tuning on continuous tokens. This leads to the same issue as the separate decoder approach, where the decoder used for generation has very poor discrete rFID.

---

> > ### Comment · Reviewer_y4uu · 2024-11-25
> >
> > Thanks for the clarification.

---

> > > ### Author Response · Authors · 2024-11-25
> > >
> > > Thank you for taking the time to review our paper and for providing constructive feedback!

---

### Official Review · Reviewer_nfS4 · 2024-11-03

**Soundness:** 3
**Presentation:** 3
**Contribution:** 2
**Rating:** 6
**Confidence:** 3

**Summary:**

This paper proposes a feasible scheme for AR model to produce high quality images effectively, and has a great improvement in performance and throughput.

**Strengths:**

1. This paper proposes a feasible scheme to solve the key problem of AR model: effective modeling of image quality, which is ahead of other methods in FID.
2. The paper is logical, the process of problem solving is very smooth, and the experimental setting is also reasonable.

**Weaknesses:**

1. There is doubt about the use of 50% for modeling discrete and continuous tokens, what happens if it is alternate, please explain the necessity of 50% here, and if there is a 30% accident, whether the other probability will break the balance, in other words, whether 50% is the best balance between the probabilities of the two modeling methods.
2. The overall solution of this paper is based on the random selection of two mature pipelines. There is a lot of randomness in training in this way, will it lead to poor stability of the network? Besides, this kind of scheme is not the most ideal for balancing the combination of continuous tokens and the separation of tokens, and it is a cheating scheme.

**Questions:**

Please explain the above questions.

---

> ### Author Response · Authors · 2024-11-23
> **Author response**
>
> **Q1: This paper is not about random selection between two mature pipelines.**
>
> HART's contributions are twofold:
>
> - Proposing a hybrid tokenizer that addresses the reconstruction quality challenge in discrete tokens;
> - Demonstrating that by predicting discrete tokens with a scalable-resolution autoregressive (AR) transformer and learning residual tokens with a compact residual diffusion model, we achieve **4.5–7.7x** higher throughput than traditional diffusion models when generating **1024px** images. This represents **the first time** AR models have shown efficiency advantages over diffusion models in high-resolution image synthesis.
>
> We would like to politely highlight that while the reviewer focused primarily on our contribution in hybrid tokenization, they **overlooked our substantial contribution to visual generation**.
>
> Below are quotes from other reviewers that recognize our significant contribution to visual generation:
>
> 1. **Reviewer 5LBN** highlights that our model architecture "exemplifies an **effective** design choice that **balances efficiency and performance**", and that "this hybrid approach not only achieves comparable speeds to VAR in class-conditional image generation tasks but also **scales effectively to large datasets**, as evidenced by its application to tasks like text-to-image generation."
>
> 2. **Reviewer yud3** mentions that our model "**demonstrates efficiency improvements over state-of-the-art** diffusion models", our residual diffusion approach "**minimizes memory and processing costs**, achieving a reduction in latency and MACs", and our method's "latency and throughput metrics are efficient, **offering a speed and computational advantage**."
>
> 3. **Reviewer y4uu** mentions that the residual diffusion idea "is **both simple and effective**" and this design "will **offer valuable insights to the community**".
>
> 4. **Reviewer a2Ay** mentions that compared with existing AR and diffusion models, HART "**achieves superior performance on text-to-image and class-conditioned image generation tasks**" and "**can generate 1024px image efficiently**".
>
> Even with our tokenization design, the choice is not a random selection between two established pipelines. Please refer to our comments below for detailed ablation studies.
>
>
> **Q2: 50%-50% sampling is a superior design choice in tokenizer training.**
>
> We perform an ablation study on the impact of different sampling ratios for continuous and discrete tokens during tokenizer training, as shown in the table below:
>
> | Discrete ratio | rFID | gFID |
> | -------- | -------- | -------- |
> | 0.0 (separate decoders) |   0.43   |  5.48   |
> | 0.1 |   0.44    |  2.65   |
> | 0.3 |   0.47    |  2.56   |
> | 0.5 |   **0.41**    |  **2.39**   |
> | 0.7 |   0.49    |  2.44   |
> | 0.9 |   0.56    |  2.51   |
> | 1.0 (discrete only) |   0.92    | 2.67    |
>
>
> As shown, a **50%-50%** sampling ratio achieves the best rFID and gFID. 30%-70% sampling proposed by the reviewer is still better than the discrete-only baseline.
>
> **Q3: HART's tokenizer training pipeline is better than alternative strategy for balancing continuous and discrete tokens.**
>
> First, we would like to clarify that our hybrid tokenizer training strategy is defintely not a *''cheating scheme''*. The decoder is trained to reconstruct original images using **both discrete and continuous tokens**, meaning there is no way to bypass learning discrete tokens. This is supported by our results in Table 1, where the discrete rFID of our hybrid tokenizer is highly competitive.
>
> Second, we experimented with an alternative design that interpolates randomly between continuous and discrete tokens during tokenizer training. This method avoids 'random selection' mentioned by the reviewer. However, the resulting generation FID is significantly worse compared to our approach.
>
> | Method | rFID | gFID |
> | -------- | -------- | -------- |
> | Interpolation-based tokenizer training |   0.49    |  2.76   |
> | HART tokenizer training |   **0.41**   |  **2.39**   |
>
> Additionally, our pipeline demonstrates strong stability. As shown in Q2, varying sampling ratios for discrete and continuous tokens during training consistently yield competitive rFID and gFID results.

---

> > ### Author Response · Authors · 2024-11-25
> >
> > Dear Reviewer nfS4,
> >
> > As the discussion period is nearing its conclusion, we kindly ask if you could review our response to ensure it addresses your concerns. Your feedback is greatly appreciated.
> >
> > Thank you for your time!
> >
> > Best,
> >
> > Authors

---

> > ### Comment · Reviewer_nfS4 · 2024-11-26
> >
> > Your answer has solved most of my doubts. I admit that 50% is the best choice at this stage, but I am also worried that such a random selection will limit further development. I look forward to your further improvement on this part, since you have fully and effectively answered my questions. I'll raise my score to 6.

---

> > > ### Author Response · Authors · 2024-11-26
> > >
> > > Dear Reviewer nfS4,
> > >
> > > Thank you for your helpful comments! Would you mind updating the main review to reflect this score change? Thank you so much and have a great day!
> > >
> > > Best,
> > >
> > > Authors

---

> > > > ### Comment · Reviewer_nfS4 · 2024-11-26
> > > >
> > > > Hi authors, I have raised my score.

---

> > > > > ### Author Response · Authors · 2024-11-26
> > > > >
> > > > > We sincerely thank the reviewer for their thorough review and valuable feedback once again!

---

### Official Review · Reviewer_yud3 · 2024-11-03

**Soundness:** 3
**Presentation:** 2
**Contribution:** 3
**Rating:** 6
**Confidence:** 5

**Summary:**

The paper presents HART, a Hybrid Autoregressive Transformer model for efficient, high-resolution text-to-image generation. Key innovations include a hybrid tokenization approach that combines discrete and continuous tokens, enabling finer detail capture while reducing computational overhead. HART achieves better efficiency, with lower latency and higher throughput than comparable models, and directly generates 1024x1024 images without super-resolution. However, the paper could benefit from a stronger theoretical foundation, exploration of alternative text conditioning methods, more comparative analysis with similar models like VAR-CLIP and STAR, and clearer differentiation from MAR, especially regarding inference optimizations. Overall, HART is a promising work in autoregressive T2I generation.

**Strengths:**

- The model demonstrates efficiency improvements over state-of-the-art diffusion models. The residual diffusion approach minimizes memory and processing costs, achieving a reduction in latency and MACs. As a result, HART’s latency and throughput metrics are efficient, offering a speed and computational advantage.

- the hybrid tokenizer combining discrete and continuous tokens is efficient in autoregressive (AR) image generation, providing higher fidelity in image reconstruction and generating fine details often missed by discrete-only models. This method addresses the usual limitations of discrete tokenizers by retaining important image details, especially at high resolutions (1024×1024）

**Weaknesses:**

- **Limited Theoretical Foundation**:
    - The paper relies predominantly on experimental results, without providing a thorough theoretical basis for the proposed methodology. A more detailed theoretical analysis would strengthen the paper’s rigor and enhance the broader applicability of the approach.
- **Ambiguity in Differentiation Between HART and other work, e.g. MAR**:
    - The distinction between HART and MAR is not fully clear, especially concerning optimizations like sampling efficiency and inference techniques. For example, while HART achieves optimal quality with just 8 sampling steps due to its diffusion setup, it is unclear why MAR could not potentially achieve similar results with similar diffusion adaptations. Additionally, some statements (lines 329-335) imply that certain optimizations, such as KV-caching, are unique to HART, but MAR could likely implement these as well. Further clarification on these points would provide a more accurate comparison between the two models and help readers understand the unique contributions of HART

- **Unexplored Alternatives for Text Conditioning**:
    - The current approach employs text tokens as the sequence start token, but it does not explore or compare this choice with other established methods, such as cross-attention mechanisms commonly used in diffusion models for text-to-image (T2I) tasks. Including comparisons with alternative conditioning methods would clarify the advantages or limitations of the current setup.
- **Insufficient Comparative Analysis**:
    - The paper would benefit from a broader evaluation against similar autoregressive T2I models, such as VAR-CLIP and STAR. A horizontal comparison would add depth to the experimental results, positioning HART more clearly within the landscape of current models.

**Questions:**

See the weaknesses, and my major concern is that:

Can you provide a more in-depth theoretical explanation of HART's hybrid tokenization and why it outperforms traditional discrete-only AR approaches? Not only to put some tricks to improve the performance. This would help clarify whether the hybrid approach could serve as a generalized framework for other AR T2I tasks.

---

> ### Author Response · Authors · 2024-11-23
> **Author response**
>
> **Q1: HART contrasts sharply with MAR.**
>
> We benchmark MAR-B with different number of sampling steps in the following table:
>
> |Sampling Step | FID | Inception Score |
> | -------- | -------- | -------- |
> | 8     | 11.18     | 184.3 |
> | 16     | 4.71     | 247.8 |
> | 32    | 4.31     | 274.0 |
> | 64    | 2.31     | 281.7 |
>
> Reducing the sampling steps in MAR results in a significant drop in both FID and inception scores. In contrast, a similar study conducted for HART-1B revealed that the number of diffusion steps has little impact on generation quality:
>
> | Sampling Step | FID | Inception Score |
> | -------- | -------- | ------- |
> | 0     | 2.23     | 312.7 |
> | 6     | 2.00     | 330.6 |
> | 8    | 2.00     | 331.3 |
> | 16    | 2.00     | 330.3 |
>
> We argue that the 'similar diffusion adaptations' mentioned by the reviewer essentially refer to decomposing image tokens into discrete and residual tokens and training a hybrid tokenizer to support this decomposition. This approach would effectively **make MAR a special case of HART**.
>
> Furthermore, KV caching cannot be applied to MAR because the authors explicitly state in their paper that MAR employs **bi-directional attention** rather than causal attention. As a result, tokens generated later modify the features of previously generated tokens, rendering KV caching inapplicable to MAR.
>
> **Q2: HART's text conditioning method is better than cross attention.**
>
> We compare cross-attention-based text conditioning with HART's text conditioning under 256x256 resolution in the table below:
>
> |Text conditiong | Parameters (M) | Throughput (img/s) | MJHQ30K FID |
> | -------- | -------- | -------- | ------- |
> | Cross attention | 317     | 14.9     | 7.51 |
> | Prepending text tokens |  **250**     | **16.0**     | **7.37** |
>
> HART's text conditioning method outperforms cross-attention, achieving better FID with 1.3x better parameter efficiency and 1.1x larger throughput.
>
> **Q3: HART is fundamentally different from STAR and VAR-CLIP.**
>
> We summarize the difference between HART and the concurrent research papers, STAR and VAR-CLIP as follows:
>
> 1. **Hybrid vs. Discrete-Only Tokenization**
>
> HART employs hybrid tokenization, where discrete tokens capture the broader structure and residual tokens represent fine-grained details. In contrast, STAR and VAR-CLIP rely solely on discrete tokens. Table 5 in our paper highlights the significant performance gains achieved by HART's residual diffusion module, which effectively models the residual continuous tokens.
>
> 2. **High-Resolution Generation**
>
> While VAR-CLIP and STAR support image generation only up to 512px resolution, HART can efficiently generate **1024px** images with **4.5–7.7x** higher throughput and comparable quality to traditional diffusion models.
>
> 3. **Text Conditioning**
>
> HART employs a unique text conditioning approach, differing from STAR (which uses cross-attention) and VAR-CLIP (which relies on the embedding of the EOS token as a single text condition token). As discussed in Q2, HART’s text conditioning method delivers superior results with 1.3x better parameter efficiency and 1.1x faster training efficiency compared to cross-attention-based methods.
>
>
> **Q4: HART has strong theoretical foundation.**
>
> We argue that HART is grounded in theoretical foundation. Hu et al. [1] (NeurIPS 2024) demonstrated that transformers are universal approximators for the score function in DiTs. When HART's diffusion head is implemented using a DiT (a special case of HART with heavier diffusion head), it can theoretically achieve universal approximation of the residual tokens under conditions in [1]. This enables HART to generate results as accurate as reconstruction outputs.
>
> In contrast, discrete-only methods are fundamentally limited to modeling categorical distributions, or the space spanned by discrete tokens. As reflected in the quantitative rFID results (Table 1) and qualitative reconstruction outcomes (Figures 4 and 14), discrete tokenizers cannot achieve as good reconstruction performance as continuous or hybrid tokenizers. Consequently, discrete-only methods are theoretically incapable of matching HART’s generation upper bound.
>
> [1] Hu et al., **On statistical rates and provably efficient criteria of latent diffusion transformers (dits)**, in NeurIPS 2024.

---

> > ### Comment · Reviewer_yud3 · 2024-11-25
> >
> > Thanks for the response. Most concerns have been solved.

---

> > > ### Author Response · Authors · 2024-11-25
> > >
> > > Thank you for spending the time reviewing our paper and providing insightful comments!

---

### Official Review · Reviewer_5LBN · 2024-11-04

**Soundness:** 3
**Presentation:** 4
**Contribution:** 3
**Rating:** 8
**Confidence:** 4

**Summary:**

This paper addresses limitations in autoregressive models (AR models), particularly the challenges in image reconstruction due to discretized tokens and the high computational costs of generating high-resolution images. To overcome these issues, the authors introduce a novel hybrid tokenizer that combines both discrete and continuous tokens. Specifically, discrete tokens are generated in an AR style, while continuous tokens are generated in a diffusion style conditioned on the discrete tokens. Compared to the VAR tokenizer, this approach demonstrates enhanced reconstruction FID performance. Moreover, in generative modeling tasks, it shows superior efficiency compared to current state-of-the-art models.

**Strengths:**

The hybrid tokenization approach presented in this paper demonstrates impressive reconstruction quality, outperforming methods that rely solely on discrete tokens. Additionally, the necessity of residual tokens is convincingly validated through detailed ablation studies, which provide strong empirical support for their inclusion. The model architecture—an autoregressive model with discrete tokens and conditioned diffusion on continuous tokens—exemplifies an effective design choice that balances efficiency and performance. This hybrid approach not only achieves comparable speeds to VAR in class-conditional image generation tasks but also scales effectively to large datasets, as evidenced by its application to tasks like text-to-image generation.

**Weaknesses:**

While Figure 7 demonstrates that residual tokens in HART are indeed easier to learn than full tokens in MAR, the results from the class-conditional image generation task seem to contradict this intuition. MAR, which generates full continuous tokens, achieves competitive FID performance despite having fewer parameters than HART, which focuses on generating residual tokens—ostensibly a less complex task.

This raises questions about the specific aspects that contribute to the observed performance gap between the two models. It would be interesting to explore whether this difference could be mitigated by increasing the diffusion steps for HART, similar to the approach in MAR. Further clarification or analysis in this area would provide valuable insights into the comparative efficiency and effectiveness of the models.

**Questions:**

In Table 4, I am interested to know if increasing the diffusion steps for HART during inference—similar to the approach used in MAR—would lead to an improvement in generation quality. Clarifying this aspect would help in understanding the impact of diffusion steps on HART's performance.

---

> ### Author Response · Authors · 2024-11-23
> **Author response**
>
> **Q1: Residual tokens are easier to learn than full image tokens.**
>
>
> (1) Most of the parameters in HART are used to model the **discrete tokens**, while only **37M** parameters are used to model the residual tokens. In fact, MAR-L uses **72M** parameters for the diffusion head, which is **1.9x more** more than ours.
>
>
> (2) We argue that runtime and MACs, rather than the number of parameters, are better measurements for modeling efficiency. HART achieves better FID and inception score than MAR with **12.9x** higher throughput and **10.9x** lower MACs.
>
> (3) The diffusion head in HART requires only **6–8 sampling steps** to achieve optimal FID and inception scores, whereas MAR typically requires **30–60 steps** (see Figure 7, left). This difference in sampling steps further demonstrates that residual tokens are inherently easier to model.
>
>
>
>
> **Q2: Clarification on Table 4 and diffusion sampling steps.**
>
>
> For clarity, we have revised Table 4. The original #steps column refers to *autoregressive sampling steps*, not *diffusion head sampling steps* for HART and MAR. In HART, the number of autoregressive sampling steps is fixed by the tokenizer's resolution hierarchy and cannot be adjusted. Below, we benchmark HART-2B with additional diffusion steps:
>
>
> Sampling Step | FID | Inception Score |
> | -------- | -------- | ------- |
> | 0     | 2.00     | 311.8 |
> | 6     | 1.79     | 329.8 |
> | 8    | 1.78     | 329.7 |
> | 16    | 1.79     | 330.5 |
>
> The residual diffusion model achieves approximately a 10% improvement in FID when the sampling steps reach at least 6. Further increasing the sampling steps yields no significant gains in quantitative performance.

---

> > ### Comment · Reviewer_5LBN · 2024-11-24
> >
> > Thanks for the added experiments and clarification, which have resolved my concerns. I have increased the score accordingly.

---

> > > ### Author Response · Authors · 2024-11-24
> > >
> > > We sincerely thank the reviewer for their positive feedback and the exceptional efforts devoted to reviewing our paper!

---

### Author Response · Authors · 2024-11-23
**General response**

We sincerely thank all the reviewers for their constructive comments and the positive feedback, including:

- **Reviewer 5LBN** mentions that HART is "**an effective design choice** that balances **efficiency** and **performance**", and our residual token design is "**convincingly validated** through **detailed ablation studies**".
- **Reviewer yud3** mentions that HART is "offering a **speed and computational advantage**" and our hybrid tokenizer provides "**higher fidelity** in image reconstruction".
- **Reviewer nfS4** mentions that our paper "proposes a feasible scheme to solve the key problem of AR model: **effective modeling of image quality**".
- **Reviewer y4uu** mentions that residual diffusion is "both **simple and effective**" and believes that our design "will offer **valuable insights** to the community".
- **Reviewer a2Ay** mentions that HART "achieves **superior performance** on text-to-image and class-conditioned image generation tasks".

We summarize our revisions (highlighted in orange) as follows:

1. **Figure 6**: We revised the figure by flipping the direction of the arrows, incorporating hidden states as conditions, and enhancing the HART attention mask for greater clarity, as suggested by Reviewer a2Ay.
2. **Section 3.3**: We made updates in alignment with the feedback provided by Reviewer a2Ay.
3. **Table 4**: We separated the number of diffusion sampling steps from the number of AR sampling steps, addressing the comments from Reviewer 5LBN.

---

### Meta-Review · Area_Chair_CZki · 2024-12-17

**Metareview:**

The paper introduces Hybrid Autoregressive Transformer (HART), a novel approach to high-resolution (1024x1024) image generation that combines hybrid tokenization – discrete tokens for structure and continuous tokens for fine-grained details – achieving state-of-the-art performance with 4.5-7.7x higher throughput and reduced computational costs compared to diffusion models. Reviewers praised the method’s strong empirical results, efficiency, and the elegant design of the hybrid tokenizer, though minor concerns were raised about the lack of theoretical depth, the fixed 50%-50% token balance, and comparative clarity with MAR, all of which the authors addressed convincingly during rebuttal. Given its clear impact, extensive experiments, and significant improvements in both reconstruction and generation quality, the paper is worthy of acceptance.

**Additional Comments On Reviewer Discussion:**

During the rebuttal period, reviewers raised key concerns regarding theoretical depth, token balance, comparative clarity, and presentation issues. Reviewer yud3 questioned the theoretical justification for hybrid tokenization and its advantage over discrete-only methods, which the authors addressed by providing comparisons and theoretical insights on residual token modeling. Reviewer nfS4 expressed doubts about the necessity of the fixed 50%-50% balance for tokenization, but the authors demonstrated through ablation studies that it achieves the best reconstruction and generation performance. Reviewer a2Ay requested clarifications on residual diffusion conditions and figure clarity (Figure 6), which the authors resolved with detailed explanations and updates to the figure. Lastly, reviewers appreciated the authors' detailed comparisons with MAR, VAR-CLIP, and STAR, which clarified HART’s unique efficiency advantages. Overall, the authors effectively addressed all concerns with empirical results, additional analyses, and clarifications, leading reviewers to resolve their doubts and in some cases, increase their scores.

---

### Decision · Program_Chairs · 2025-01-22

Accept (Poster)